# Batch and Continuous Lipase-Catalyzed Production of Dietetic Structured Lipids from Milk Thistle, Grapeseed, and Apricot Kernel Oils

**DOI:** 10.3390/molecules30091943

**Published:** 2025-04-27

**Authors:** Şuheda Akbaş, Natália M. Osório, Suzana Ferreira-Dias

**Affiliations:** 1Department of Food Engineering, Faculty of Engineering, Talas Campus, Erciyes University, 38039 Kayseri, Türkiye; suhedakb7@gmail.com; 2LEAF—Linking Landscape, Environment, Agriculture and Food Research Center, Instituto Superior de Agronomia, Universidade de Lisboa, 1349-017 Lisbon, Portugal; 3MARE—Marine and Environmental Sciences Centre, Escola Superior de Tecnologia do Barreiro, Instituto Politécnico de Setúbal, 2839-001 Lavradio, Portugal; natalia.osorio@estbarreiro.ips.pt; 4Laboratório de Estudos Técnicos, Instituto Superior de Agronomia, Universidade de Lisboa, 1349-017 Lisbon, Portugal

**Keywords:** acidolysis, apricot kernel oil, batch bioreactor, continuous bioreactor, grapeseed oil, interesterification, lipase, low-calorie structured lipids, milk thistle oil, structured lipids

## Abstract

The sustainable production of healthy structured lipids (SLs) using oils extracted from agro-industry by-products or non-conventional lipid sources is of utmost importance in the framework of a circular bioeconomy, toward a zero-waste goal. In this study, low-calorie triacylglycerols (TAGs) containing a long-chain (L) fatty acid (FA) at position *sn*-2 and medium-chain (M) FAs at positions *sn*-1,3 (MLM type SL) were obtained from virgin cold-pressed milk thistle (51.55% linoleic acid; C18:2), grapeseed (66.62% C18:2), and apricot kernel (68.61% oleic acid; C18:1) oils. Lipase-catalyzed acidolysis with capric acid (C10:0) or interesterification with ethyl caprate (C10 Ethyl) in solvent-free media were performed. In batch reactions, immobilized *Rhizomucor miehei* lipase (Lipozyme RM) was used as a biocatalyst. For all tested oils, new TAG (SL) yields, varying from 61 to 63%, were obtained after 6 h of interesterification. Maximum new TAG yields were reached after 6, 24, and 30 h of acidolysis with grapeseed (64.7%), milk thistle (56.1%), or apricot kernel (69.7%) oils, respectively. Continuous acidolysis and interesterification of grapeseed oil were implemented in a packed-bed bioreactor, catalyzed by immobilized *Thermomyces lanuginosus* lipase (Lipozyme TL IM). Throughout 150 h of continuous operation, no lipase deactivation was observed, with average SL yields of 79.2% ± 4.1 by interesterification and 61.5% ± 5.91 by acidolysis.

## 1. Introduction

The search for nutritionally healthy lipids is a growing concern for consumers and for the food industry. Moreover, using oils obtained from non-conventional seeds of wild plants or medicinal plants, or from agro-industrial residues extracted by mechanical processes, avoiding extraction by pollutant organic solvents, is a trend that meets the requirements of the circular bioeconomy toward a zero-waste situation.

The importance of using oils from agro-industry residues for human consumption, namely the grapeseed oil extracted from the seeds of grapes (*Vitis vinifera* L.) after winemaking and the oil extracted from rice bran (*Oryza sativa* L.), was recognized by the FAO, in the Codex Alimentarius standard for named vegetable oils, in the documents from 1999 [1] and 2015 [2], respectively. In 2022 [3], nut oils specifically from almonds (*Amygdalus communis* L.), pistachios (*Pistacia vera* L.), walnuts (*Juglans regia* L.), and hazelnuts (*Corylus avellana* L.) were added to the list, showing the importance of the high nutritional quality of these oils (rich in oleic and/or linoleic acids).

Grapeseed oil is a winemaking by-product of interest owing to its composition. Oil constitutes around 6–20% of grape seeds and its chemical composition depends mainly on the degree of maturation of the seeds employed, grape variety, and environmental cultivation conditions, and to a lesser degree, on the seed extraction methodology [4]. Grapeseed oil is also noted for its beneficial nutritional value due to its richness in unsaturated fatty acids (higher than sunflower, safflower, and corn oils), especially linoleic (C18:2) acid (58–78%, *w*/*w*) and oleic (C18:1) acid (12–28%, *w*/*w*). It also contains stearic (C18:0) and palmitic (C16:0) acids, which are present in lower amounts (3–6.5%, and 5.5–11%, *w*/*w*, respectively) [3,4].

The oil extracted from apricot (*Prunus armeniaca* L.) kernels, an agro-industrial by-product, is utilized worldwide for edible, cosmetic, and medical purposes. Apricot kernel oil is rich in unsaturated fatty acids, particularly oleic and linoleic acids, which are known for their health benefits. The typical fatty acid composition includes oleic acid (C18:1), constituting approximately 58% to 70.7%, linoleic acid (C18:2), which accounts for about 20.5% to 29%, and palmitic acid (C16:0), ranging from 3.1% to 7.8%. Stearic acid (C18:0), is present at approximately 1.4%, and linolenic acid (C18:3) is found in smaller amounts of around 0.4% to 1.4% [5,6,7].

Milk thistle (*Silybum marianum* L. Gaert.) is an annual or biennial plant, a member of the *Asteraceae* family. It is native to some parts of Europe, Africa, and Asia and is now widespread all over the world. It grows as a weed on roadsides or in empty fields. Milk thistle is also grown specifically as a medicinal plant, since its seeds contain a biologically active compound called silymarin, with recognized medical applications [8,9]. Dried achenes of *Silybum marianum* contain 18–31% oil [10,11,12]. This oil is particularly rich in unsaturated fatty acids, with linoleic acid being the most abundant (42–54%), followed by oleic acid (23–36%). Palmitic acid is present in amounts of 7–8%. Additionally, smaller quantities of linolenic acid, stearic acid, and other fatty acids have been identified. Milk thistle oil also contains notable levels of phospholipids, phytosterols, and vitamin E, making it suitable for food applications [10,11,12].

In addition to the natural biological value of several edible oils, their functional properties can be improved by lipase-catalyzed modification to obtain so-called structured lipids (SLs). SLs are obtained by modifying natural triacylglycerols (TAGs) or phospholipids through (i) the incorporation of new fatty acids (FAs) at positions *sn*-1,3 of the TAGs and keeping the original unsaturated/polyunsaturated fatty acid at position *sn*-2, or (ii) by changing the distribution of fatty acids in the glycerol backbone [13,14,15,16,17,18,19]. Among the SLs obtained by TAG restructuring, low-calorie TAGs are important products for dietetic and medical applications. These dietetic structured lipids are triacylglycerols containing medium-chain fatty acids (M) at positions *sn*-1 and *sn*-3, and a long-chain fatty acid (L) at position *sn*-2 (MLM). These structured TAGs have an energy content ranging from 5 to 7 kcal/g, whereas conventional fats and oils provide 9 kcal/g. In vegetable oils, mono- and polyunsaturated long-chain fatty acids, including essential fatty acids, are preferentially located at position *sn*-2. Medium-chain fatty acids have a lower caloric value than long-chain FAs and are metabolized in the liver, like glucose, rather than being stored as fat [15,19,20]. Dietetic MLM-type structured lipids are currently synthesized by (i) acidolysis with medium-chain fatty acids, namely caprylic (C8:0) or capric (C10:0) acids; or (ii) interesterification with ethyl octanoate (C8 Ethyl) or ethyl decanoate (C10 Ethyl), using *sn*-1,3 regioselective lipases (E.C. 3.1.1.3., triacylglycerol acylhydrolases) as biocatalysts [21,22,23,24,25,26,27,28,29,30,31,32,33,34,35]. These SLs cannot be synthesized using chemical catalysts because they are not regioselective and act at random in TAG backbone. Moreover, enzymatic synthesis of structured lipids occurs under milder reaction conditions (e.g., temperature, pressure, absence of toxic organic solvents) than chemical catalysis, preserving SL from thermal oxidation, avoiding side-reactions, as well as allowing easier product recovery, among others [13,14,15,16,17,18,19,36].

The synthesis of dietetic low-calorie SLs by immobilized enzymes in solid carriers can be performed in batch or in continuous bioreactors, either in the presence of an organic solvent or in solvent-free media [23,26,27,28,29,30,31,32,34,35,37,38].

Batch stirred tank reactors (STRs) are preferred due to their ability to achieve a high degree of substrate dispersion and their ease of reaction control. However, they have drawbacks, including high shear forces on the support, time-consuming cleaning processes, and high energy consumption [39]. In contrast, packed-bed reactors (PBR) are advantageous as they exhibit low shear due to the lack of mechanical agitation, resulting in enhanced catalyst stability. Both batch PBRs with continuous recycling of reaction medium [28] and continuous PBRs [21,22,33,40] or batch fluidized packed bed reactors with recirculation [41] have been used for MLM production, in media formed by substrate blends.

Conventional edible oils, like olive oil [23,42], rapeseed and safflower oils [21], fish oil [22], soybean oil [28], sesame oil [43], peanut oil [44], rice bran oil [45], pumpkin oil [46,47], cottonseed oil [41], and coconut oil [48], have been used for low-calorie SL production. Moreover, the use of non-conventional oils extracted from agro-industry by-products, namely avocado oil [24], grapeseed oil [25,27,33,49], spent coffee ground oil [29,30], and crude olive pomace oil [29,34], or non-conventional edible oils, such as argan oil [32], sacha inchi oil (*Plukenetia volubilis* L.) [50], and microbial oils [26,51], have also been reported.

In this study, cold-pressed virgin oils extracted from agro-industry by-products, namely from grape seeds or apricot kernels, and from milk thistle seeds, were used for the synthesis of low-calorie MLM-type structured lipids, by batch acidolysis with capric acid (C10:0) or interesterification with its ethyl ester (ethyl caprate/ethyl decanoate), catalyzed by immobilized *sn*-1,3 regioselective *Rhizomucor miehei* lipase. Acidolysis and interesterification of grapeseed oil were also implemented in a continuous packed-bed reactor, using immobilized *Thermomyces lanuginosus sn*-1,3 regioselective lipase as a catalyst.

## 2. Results and Discussion

### 2.1. Oil Characterization

The quality parameters, namely the acid value (AV) and peroxide value (PV), of the three oils are shown in Table 1. All oils are of high quality since both the AV and PV are below the limits for cold-pressed and virgin oils presented by the Codex Alimentarius [3]: AV of 4.0 mg KOH/g oil and PV of 15 milliequivalents of active oxygen per kilogram of oil.

Table 2 and Table 3 show the fatty acid and TAG compositions of the oils used in this study. Linoleic acid is the major fatty acid of grapeseed (66.62%) and milk thistle (51.55%) oils, followed by oleic (19.96 and 28.02%, respectively) and palmitic (7.01 and 7.71%, respectively) acids. In apricot kernel oil, oleic acid is the major fatty acid (68.61%), followed by linoleic (23.89%) and palmitic (5.16%) acids. Polyunsaturated fatty acids (PUFAs) account for 67.76, 51.76, and 24.01%, while monounsaturated fatty acids (MUFAs) correspond to 20.42, 29.23, and 69.50% of the total fatty acids in grapeseed, milk thistle, and apricot oils, respectively.

The fatty acid composition of grapeseed oil is within the values reported for this oil in the Codex Alimentarius standard for named vegetable oils [3].

Wu et al. [52] extracted the kernel oil from youyi sweet apricot cultivar, grown in China, by mechanical pressing at different temperatures (40–200 °C). The obtained oils presented the following variation range for the major fatty acids: C18:1 (74.25–75.20%); C18:2 (18.47–19.39%); and C16:0 (4.16–4.23%). Saturated fatty acids (SFAs) varied from 5.27 to 5.36%; MUFAs varied from 75.17 to 76.13; and PUFAs ranged from 18.55 to 19.47% [52].

Gupta et al. [53] evaluated the fatty acid composition of wild apricot kernel oils. The fatty acid profile of these oils showed that the contents of oleic acid varied from 62.07 to 70.6%; linoleic acid from 20.5 to 27.76%; linolenic acid ranged from 0.4 to 1.42%; and palmitic acid varied from 5.0 to 7.79%. The apricot kernel oil used in our study showed a fatty acid composition similar to that of the samples of wild apricot kernel oil evaluated by Gupta et al. [53], while slight differences were observed with respect to the sweet apricot kernel oil evaluated by Wu et al. [52].

Milk thistle seed oils from plants grown in China presented linoleic (45.83–46.41%), oleic (30.12–30.59%), palmitic (7.87–8.19%), and stearic (6.59–6.69%) acids as the major fatty acids [12]. Turkish milk thistle oil presented 51.97% linoleic, 27.06% oleic, 5.33% stearic, and 10.39% palmitic acids [9]. The milk thistle oil used in the present study has a fatty acid composition more like that of the oil analyzed by Ayduğan et al. [9].

Table 3 shows that all the oils presented TAGs with an equivalent carbon number (ECN) ranging from 42 to 52. In grapeseed oil, the majority of TAGs (c.a. 72%) have an ECN equal to 42 and 44. In milk thistle oil, the most abundant TAGs have an ECN of 44 (31.5%), followed by TAGs with ECNs of 46 (26.82%), 48 (19.15%), and 42 (15.2%). Apricot kernel oil presents 43.5% of TAGs with ECN 48, followed by 35.6% TAGs with an ECN of 46 and 15.8% of TAGs with an ECN of 44.

The TAG profile of grapeseed oil is in good agreement with the fatty acid composition of grapeseed oil (Table 2), since the major TAGs contain at least two molecules of linoleic acid (ECN = 42 and ECN = 44) (Table 3).

Milk thistle oil, having a fatty acid composition (51.6% linoleic acid and 28.0% oleic acid) similar to that of grapeseed oil (66.6% linoleic acid and 20.0% oleic acid), presents a similar TAG profile. However, TAGs with ECN 42 only represent 15% of total TAGs, while TAGs with ECN 46 and ECN 48 represent about 27 and 19% of the total TAGs, against 20 and 7% in grapeseed oil, respectively.

Apricot kernel oil contains 68.6% oleic acid and 23.9% linoleic acid (Table 2). Therefore, its major TAG is triolein (OOO; ECN 48), representing about 35% of total TAGs, followed by about 41% TAGs with ECN 44 and 46, containing one or two oleic acids in their molecules (OOL, OLL + OOLn + PoOL).

The TAG composition of grapeseed oils extracted from 21 samples of red grape and 11 samples of white grape varieties grown in Italy was evaluated by De Marchi et al. [54] using MALDI-TOF and ESI/MS techniques. Triolein was the major TAG, with an average of 42.8%, ranging from 33.5 to 51.6%, followed by LOL (average = 23.3%; 19.0–26.4%); LPL (average = 15.1%; 12.3–17.9%), LSL + LOO (average = 11.1%; 7.7–15.9%), and LOP (average = 6.1%; 4.3–9.1%). Our grapeseed oil sample had a TAG profile within the values observed by De Marchi et al. [54].

Other studies on milk thistle oil confirmed that this oil mainly contains five major triacylglycerols rich in linoleic acyls, namely LLL, LLO, LLP, LOO, and LOP [11,12]. Zhang et al. [12] evaluated the TAG composition of Chinese milk thistle oils. The most abundant triacylglycerol in these oils was OLL (~20–21%), followed by LLL (~18%), POL (~15%), and PLL (~11%). Our results are not very different from those found in the milk thistle oils evaluated by Zhang et al. [12].

The TAG composition of kernel oil from the youyi sweet apricot cultivar, grown in China, was evaluated by Wu et al. [52]. All samples showed similar TAG profiles comprising 20 different types. However, the content of TAGs varied with pressing temperatures. The predominant TAGs were POO (~7–18%), POL (~2–11%), SOO (~1–9%), OLL (~9–12%), OOL (~10–14%), and LLL (~8–16%). The apricot kernel oil used in our study had a higher OOL content (27.8%) and lower LLL content (2.5%), while the other TAG species presented similar values to those of the oils evaluated by Wu et al. [52]. These differences may be ascribed to the cultivar and/or to the mechanical extraction conditions used, which we do not know, since commercial apricot oil was used.

Different cultivars, extraction methods, and edaphoclimatic conditions may explain the variations observed between the fatty acid and TAG composition of the virgin oils used in our study and of those reported in the literature.

### 2.2. Batch Production of Low-Calorie TAGs by Acidolysis and Interesterification

Acidolysis consists of a reaction between an ester (TAG) and free fatty acids, which will replace the original fatty acids in the TAG molecules. When *sn*-1,3 regioselective lipases are used, this exchange will occur at positions *sn*-1,3. The interesterification consists of an ester–ester interchange being, in the present study, between the oil TAGs and capric acid ethyl ester.

Both acidolysis and interesterification reactions are usually considered as a two-step reaction, consisting of the hydrolysis of ester bonds in the TAGs followed by the esterification of acyl groups in the glycerol backbone. The hydrolysis of a TAG results in a diacylglycerol (DAG) and the release of a fatty acid molecule. The formed DAG can (i) esterify with another free fatty acid (FFA) to form a new TAG or (ii) be hydrolyzed into a monoacylglycerol (MAG) and a fatty acid molecule. Diacylglycerols and monoacylglycerols are thermodynamically unstable compounds, acting as reaction intermediates [55].

The rate of DAG formation increases with the water content in the system [55]. For the hydrolytic step, the presence of water is required. According to Heisler et al. [56], in the isomerization of 1,2-dipalmitin into the 1,3-isomer, the trace water present in lyophilized *sn*-1,3 regioselective lipase powder was used in the first step of DAG hydrolyses. These authors also observed that the hydrolytic step was faster than the esterification step.

Therefore, the control of the water content in the system is very important since, to achieve high yields of new TAGs, low contents of DAGs, MAGs, and FFAs are required at the equilibrium. This means that, after the hydrolytic initial step, the esterification reaction must occur at a higher rate than hydrolysis. The reaction equilibrium depends on the extent of hydrolysis versus the esterification step and the required optimal amount of water mainly depends on the type of lipase used [55].

For the three oils used, batch acidolysis with capric acid and interesterification with ethyl caprate (ethyl decanoate), C10 ethyl, catalyzed by immobilized *Rhizomucor miehei sn*-1,3 regioselective lipase (Lipozyme RM), was performed in solvent-free media for 48 h at 40 °C. A biocatalyst load of 5% (*w*/*w*) of the amount of triacylglycerols and a molar ratio of oil/acyl donor of 1:2, corresponding to the stoichiometric ratio for *sn*-1,3 regioselective lipases were used.

The time courses of the acidolysis and interesterification reactions are presented in Figure 1 and Figure 2, respectively. Along the reactions, initial TAGs were converted into other species (new TAGs, diacylglycerols, DAGs, or monoacylglycerols, MAGs) because both acidolysis and interesterification are sequential reactions. When regioselective *sn*-1,3 lipases are used, the fatty acids at positions *sn*-1,3 in the original TAGs are released to the reaction medium, increasing the FFA contents in the reaction media and leading to the production of partial acylglycerols (DAGs and MAGs). Afterward, the esterification of capric acid to *sn*-1,3 ester bonds occurs with the synthesis of new TAGs of MLL and MLM types [22]. These new TAGs are dietetic low-calorie SL.

The conversions of the initial TAGs into the new species and the yields in new TAGs were calculated (c.f. 3.5). The results concerning acidolysis and interesterification are presented in Figure 3 and Figure 4, respectively.

As expected, the conversion values increased during the 48-h time course and were always higher than the yield of new TAGs. Except for the acidolysis of apricot kernel oil, where the new TAG yield reached a plateau after 24 h of reaction, a decrease in the new TAG yield was observed after reaching a maximum yield value. Either in acidolysis or in interesterification, when the hydrolysis of an ester bond in TAG is faster than the esterification step, a decrease in the yield of new TAGs is observed [55]. The large differences between TAG conversion and the new TAG yield suggest that the hydrolytic step is faster than the esterification reaction, which results in lower production of new TAGs and increases in MAGs, DAGs, and FFAs in the reaction medium. These differences are particularly important in the acidolysis of milk thistle oil, throughout the reaction time course (Figure 3), and in all interesterification reactions after attaining the maximum yield (Figure 4).

Table 4 shows the maximum yields of new TAGs, maximum conversion values, the respective time to attain these values, as well as the initial rates of new TAG formation for all the systems studied. When grapeseed oil was used, maximum yields were attained very quickly, either by acidolysis or interesterification (after 6 h reaction), and the values (64.7% by acidolysis and 61.0% by interesterification) were not very different. However, the initial rate of interesterification was 3.3-fold the acidolysis rate. The same behavior was observed with milk thistle and apricot kernel oils: the interesterification reaction rates were 4.4 and 3.6-fold the acidolysis rates, respectively. In fact, the times to attain the maximum new TAG production were 24 and 30 h for the acidolysis with milk thistle and apricot kernel oils, respectively, against 6 h for the interesterification of both oils with ethyl caprate. The maximum yields of low-calorie TAGs varied from 56.1% (milk thistle acidolysis) to 69.7% (apricot kernel oil acidolysis). Although the acidolysis rate was lower, the maximum yields did not depend on the reaction chosen.

The TAG conversion to other species increased throughout the 48 h reactions and the values attained varied from 85.2% and 85.7% (apricot kernel oil acidolysis and interesterification, respectively) to 90.0% by acidolysis, and 92.2% by the interesterification of grapeseed oil. These results suggest a slight preference of Lipozyme RM toward oils richer in linoleic acid (66.62% in grapeseed oil and 51.55% in milk thistle oil) than in oils richer in oleic acid (68.61% C18:1 and 23.89% C18:2 in apricot kernel oil).

For the samples of each oil with the highest new TAG yield (Table 4), the TAG fraction was analyzed by HPLC to identify the different TAG species. Figure 5 shows the TAG profile (grouped by ECN) of the initial oils and of the modified oils by acidolysis or interesterification, when the highest new TAG yields were attained. As expected, the new TAG fraction corresponds to TAGs with ECN smaller than 42, accounting for 75.2% (in modified grapeseed oil by acidolysis) to 97.0% (in modified grapeseed oil by interesterification) of the total TAGs. The main new TAGs, containing one or two capric acid, C, residues (mainly at positions *sn*-1,3, but some at position *sn*-2 due to eventual acyl migration phenomenon), will be: CLC (ECN = 34); COC (ECN = 36); LLC (ECN = 38); OLC (ECN = 40); and OOC (ECN = 42).

In most of the studies on the synthesis of low-calorie MLM-type TAGs, the reaction extent has been evaluated by the incorporation degree of medium-chain fatty acids in the original TAGs. In our study, since the reaction kinetics were followed by the quantification of initial TAGs and new TAGs, DAGs, MAGs, and FFA in the reaction medium, comparison with several results in the literature is not always straightforward.

In batch acidolysis of arachidonic acid-rich microbial oil from *Mortierella alpina* with caprylic acid in a solvent-free system using Lipozyme RM IM, a substrate molar ratio oil:caprylic acid equal to 1:3, and a temperature of 60 °C, the highest incorporation of caprylic acid (20.14%) was attained after 6 h of acidolysis [26].

Van Nguyen and Shahidi [48] carried out the acidolysis of virgin coconut fat with omega-3 PUFA (eicosapentaenoic, EPA, or docosahexaenoic, DHA, acid, separately, or in blends of EPA + DHA), in hexane media. Immobilized lipases Lipozyme TL IM, Lipozyme IM60 (lipase from *R. miehei*), and lyophilized non-regioselective *Candida rugosa* lipase were tested. The highest incorporations of EPA, DHA or EPA + DHA, were obtained with Lipozyme TL IM. The optimized reaction conditions were found by response surface methodology, as follows: 3.3% biocatalyst load, 42 °C, and 33.4 h. Under these conditions, omega-3 PUFA incorporation in coconut oil was around 33% for DHA, 45% for EPA, and 47% for the blend of both fatty acids.

The acidolysis of extra-virgin olive oil with caprylic or capric acids was carried out either in solvent-free or in hexane media, catalyzed by Lipozyme RM IM, by the immobilized lipase from *Thermomyces lanuginosus* (Lipozyme TL IM), or by immobilized *Candida antarctica* lipase (Novozym 435), for 24 h at 45 °C, using the stoichiometric molar ratio of olive oil:fatty acid of 1:2 [23]. For all tested lipases, higher incorporations were obtained for capric acid, ranging from 27.1 to 30.4 mol%, in solvent-free media [23].

To make the comparison of our results with similar studies easier, some examples of batch acidolysis and interesterification of several oils with capric acid or C10 Ethyl, carried out in solvent-free media, with different biocatalysts, as well as the best reaction conditions and results, are shown in Table 5.

Grapeseed oil was previously used to produce MLM by acidolysis in solvent-free media, with caprylic or capric acids, using a molar ratio for oil:fatty acid of 1:2 [25,27] or varying the molar ratio from 1:1 to 1:3 [49]. Non-commercial *sn*-1,3 regioselective lipases, namely a recombinant *Rizopus oryzae* lipase immobilized in a resin (rROL), and *Carica papaya* lipase (CPL), self-immobilized in papaya latex, were used by Costa et al. [31]. The yields in new TAGs were 68.5% with C8:0, and 52.4% with C10:0, after 24 h of acidolysis catalyzed by rROL, and 40.8% and 38.2%, after 48 h acidolysis with caprylic or capric acid catalyzed by CPL, respectively [25].

Bassan et al. [27] used the commercial immobilized lipases Lipozyme RM IM, Lipozyme TL IM, and Novozym 435, as catalysts, reaching incorporation degrees varying from 23.62 to 34.53 mol%, after 24 h of acidolysis at 45 °C. The highest incorporation was obtained using Lipozyme RM IM in the acidolysis of grapeseed oil with capric acid (34.53 ± 0.05 mol%).

Liu and Akoh [49] tested Lipozyme RM IM and Lipozyme 435 (*Candida antarctica* recombinant immobilized lipase) as catalysts for the acidolysis of grapeseed oil with capric acid, following a Taguchi experimental design, to optimize capric acid incorporation. For both biocatalysts, the best reaction conditions are presented in Table 5. New TAGs contained 46.0 mol% and 47.3 mol% at the *sn*-1,3 positions when Lipozyme RM IM and Lipozyme 435 were used, respectively. Since acyl migration occurred, the total incorporation of capric acid in new TAG species was 60.1 mol% with Lipozyme RM IM and 50.8 mol% with Lipozyme 435.

In batch reactions, Lipozyme TL IM preferred interesterification with C10 ethyl than acidolysis with capric acid, when using crude spent coffee ground (SCG) and crude olive pomace (OP) oils, in solvent-free medium (Table 5) [29]. However, when *Rhizopus oryzae* lipase immobilized in magnetic nanoparticles (ROL-MNP) was used, the acidolysis reaction was faster than interesterification (Table 5) [29].

Mota et al. [30] used crude SCG oil to produce MLM, either by acidolysis with caprylic acid or capric acid, or by interesterification with ethyl caprylate (C8 ethyl) or C10 ethyl, catalyzed by Lipozyme RM IM or Lipozyme TL IM (Table 5). With this oil, higher yields in new TAGs were obtained by acidolysis when Lipozyme RM IM was used, or by interesterification when Lipozyme TL IM was used [30].

In the present study, where Lipozyme RM was used in batch reactions, interesterification was faster than acidolysis, but the new TAG yield was more dependent on the oil used than on the reaction followed.

Concerning biocatalyst load, only 5% of the weight of the oil was used in our study. Moreover, the molar ratio corresponded to the stoichiometric value when *sn*-1,3 regioselective lipases are used, and both acidolysis and interesterification reactions were performed at 40 °C, which resulted in lower operation costs (less enzyme, substrates, and energy costs).

In the examples of Table 5, enzyme load was, in most cases, equal to 5% (*w*/*w*), except for the acidolysis of sacha inchi oil, where 10% of *Rhizopus oryzae* lipase immobilized in corn cob powder was used [50]. The molar ratios of oil/C10:0 or oil:C10 ethyl were usually 1:2 (stoichiometric value) or 1:3 (slight excess of acyl donor), except in the case of the acidolysis of pumpkin oil, catalyzed by Lipozyme RM IM, where a large excess of C10:0 (molar ratio 1:9) was used [47]. This large excess of capric acid amounted to 48.4 mol % incorporation [47]. Using Lipozyme RM IM and a molar ratio of 1:2 in the acidolysis of olive oil or grapeseed oil, the incorporation was 27.1 mol % [23] and 34.5 mol %, respectively [27]. For a molar ratio of 1:3, 60.08 mol% incorporation of C10 in grapeseed oil was observed [49], which was higher than the value reported by Atsakou et al. for a molar ratio of 1:9 [47]. In fact, the use of a large excess of capric acid seems not to be directly related with its incorporation in TAGs because a possible inactivation of the biocatalyst may occur due to the presence of large amounts of medium-chain fatty acids in reaction media. Moreover, costs related to unreacted capric acid recovery and product purification will increase.

In the present study, the new TAG yields obtained by acidolysis, catalyzed by Lipozyme RM (Table 4), were higher than (i) the values obtained with grapeseed oil when immobilized rROL and *Carica papaya* lipase were used [25]; (ii) the values obtained with crude olive pomace (OP) oil and crude spent coffee grounds (SCG) oil catalyzed by Lipozyme TL IM or by ROL-MNP; but (iii) similar to the incorporation in SCG oil catalyzed by Lipozyme RM IM [30]. Concerning batch interesterification, the time to attain a pseudo-equilibrium (6 h in our study) was similar to that observed with crude olive pomace oil and crude spent coffee grounds oil (7 h) [29,30]. Moreover, the new TAG yields were higher than the values obtained by interesterification of crude olive pomace oil (52%), but similar to the value obtained with crude spent coffee grounds oil (65%) catalyzed by Lipozyme TL IM [29], and lower than the value observed with crude spent coffee grounds oil when Lipozyme RM IM was used (~70%) [30].

The various studies on the lipase-catalyzed production of low-calorie SL of MLM type showed that the preference of a specific biocatalyst for acidolysis or interesterification depends on the oil and operation conditions used.

### 2.3. Continuous Production of Low-Calorie TAGs

Since the highest rates of batch acidolysis and interesterification reactions were observed with grapeseed oil, this oil was selected for reaction implementation in a continuous bioreactor. High reaction rates are important in continuous process implementation because they allow for shorter residence times to attain high conversions and yields. A continuous packed-bed bioreactor with up-flow feeding, using the same reaction medium and temperature (40 °C) as in batch reactions, was used. Instead of Lipozyme RM, Lipozyme TL IM was chosen. This biocatalyst is very sensitive to shear stress, which makes it unsuitable for batch reactions under magnetic stirring, but has a much lower price than Lipozyme RM (about eight-fold cheaper, according to the manufacturer), contributing to the economic viability of the enzymatic processes.

The bioreactor operated continuously for 150 h in acidolysis and 160 h in interesterification. Figure 6 shows the continuous production of new TAGs, the residual TAGs, and the ratio between the amount of new TAGs and the total TAGs (new TAG yield). The biocatalyst maintained its activity in both acidolysis and interesterification throughout the continuous operation. However, the average yield of new TAGs was higher by interesterification (78% ± 4.1) than by acidolysis (62% ± 5.9).

The preference of Lipozyme TL IM for interesterification over acidolysis was also observed in the batch production of low-calorie TAGs from spent coffee ground oil [36].

The operational stability of immobilized lipases during the synthesis of low-calorie structured lipids has been evaluated more frequently in consecutive batch reuses than in continuous operation.

In batch acidolysis of virgin olive oil with capric acid, Lipozyme TL IM showed first-order deactivation kinetics (47.2 h) [23]. When used in a packed-bed reactor with continuous recirculation for the interesterification of soybean oil with TAGs rich in caprylic and capric acids, no significant deactivation of this biocatalyst was observed after 25 consecutive batches (total of 6.7 h) [28]. The operational stability exhibited by Lipozyme TL IM, in batch acidolysis reuses, was lower than that observed for this biocatalyst in the present study using a continuous PBR.

In continuous interesterification in a packed bed reactor, between fish oil and medium-chain TAGs, no deactivation of Lipozyme TL IM was observed throughout a two-week continuous operation [22].

Souza-Gonçalves et al. [40] performed the acidolysis of highly acidic (12–29% FFA) crude olive pomace oils with C8:0 or C10:0 acids and interesterification with their ethyl ester forms, catalyzed by Lipozyme TL IM or Lipozyme RM IM, in continuous PBR for 70 h in solvent-free media at 40 °C. Throughout the continuous operation, no biocatalyst deactivation was observed, except for Lipozyme TL IM in the acidolysis with capric acid (linear deactivation; half-life time = 228 h) and Lipozyme RM IM in the interesterification with ethyl caprylate (first-order deactivation; half-life time = 74 h).

The high stability of Lipozyme TL IM observed in our study, either in the acidolysis of grapeseed oil with capric acid or interesterification with C10 ethyl, is comparable to the results previously obtained with this biocatalyst when used in continuous bioreactors [22,40].

## 3. Materials and Methods

### 3.1. Materials

Cold-pressed organic milk thistle oil was obtained from HerbalNordPol-Gdansk Sp, Pomerania, Poland; cold-pressed organic apricot kernel oil was obtained from AROMA Labs, İstanbul, Turkey, and cold-pressed grapeseed oil was produced by Destilaria Levira, Lda., Anadia, Portugal. The commercial immobilized *sn*-1,3 regioselective lipase Lipozyme RM (*Rhizomucor miehei* lipase immobilized on a resin carrier) and Lipozyme TL IM (lipase from *Thermomyces lanuginosus* immobilized on a granular resin, Immobead 150) were a gift from Novonesis, Lyngby, Denmark; capric acid (C10:0; >98% purity) and ethyl caprate (C10 ethyl; >98% purity) were provided by TCI Europe N.V., Zwijndrecht, Belgium. All reagents used were of analytical-grade.

### 3.2. Oil Characterization

Oil samples were characterized with respect to the following properties: acidity expressed as acid value, peroxide value, and fatty acid (FA) and TAG composition. Both the oil acidity and peroxide value were determined in triplicate in accordance with the methodology of the International Olive Council, IOC [57,58]. The fatty acid composition was evaluated by gas chromatography after their FA conversion into fatty acid methyl esters (FAME) following the recommended methodology of the IOC [59]. A Perkin Elmer Autosystem 9000 gas chromatograph (GC) (Perkin Elmer, Waltham, MA, USA), equipped with FID and a fused silica capillary column Supelco SPTM—2380 (60 m × 0.25 mm × 0.2 μm film thickness) was used. The injector and FID were set at 250 and 260 °C, respectively. The column temperature was set at 165 °C for 45 min, increasing at a rate of 7.5 °C/min up to 230 °C, and holding for 25 min at this temperature. Helium was used as the carrier gas at a pressure of 20.0 Psig. FAME standards (GLC-10 FAME mix, 1891-1AMP, from Sigma-Aldrich, St. Louis, MO, USA), analyzed under the same conditions, were used for fatty acid identification.

The TAG profile was evaluated by HPLC according to the method of the International Olive Council [60]. The HPLC chromatograph consisted of a Perkin Elmer binary LC pump 250 YL9170 (Perkin Elmer, Waltham, MA, USA), a refractive index detector, and a HPLC column (25 cm × 4 mm i.d.), packed with RP-18 phase (4 μm particle size) (Lichrosorb (Merck, Rahway, NJ, USA) RP 18 Art 50333 column), placed in a thermostatic oven to maintain temperatures between 15 and 20 °C (Altech 530 heater, Nicholasville, Kentucky, USA) Propionitrile, super-purity or HPLC-grade, was used as the mobile phase at a flow rate of 0.6 mL/min. TAGs were separated according to their equivalent carbon number (ECN), given by the number of carbons present in the fatty acid chains minus the number of double bonds in TAG fatty acids, multiplied by two.

### 3.3. Lipase-Catalyzed Batch Reactions

The production of low-calorie structured lipids was carried out by acidolysis or interesterification in 20 mL cylindrical jacketed glass reactors for water circulation to maintain the temperature at 40 °C under magnetic stirring (300 rpm) and closed with stoppers. The reaction media consisted of blends of virgin oils (10 g of grapeseed, apricot kernel or milk thistle oils), with capric acid (C10:0) in the acidolysis reaction or ethyl caprate (C10 Ethyl) in the interesterification reaction. A molar ratio of oil/acyl donor of 1:2, corresponding to the stoichiometric ratio, was used. When the reaction temperature was attained, the biocatalyst (Lipozyme RM) was added to the reaction medium at a load of 5% (*w*/*w*) of the amount of triacylglycerols. Samples (0.5 mL) were collected throughout the 48 h time course and immediately frozen at −18 °C until analysis. Reactions were performed in duplicate.

### 3.4. Lipase-Catalyzed Continuous Reactions

The continuous bioreactor consisted of a double-jacket glass column (height = 20 cm; internal diameter = 2 cm) for water recirculation for temperature control, with a glass G0 sieve on the bottom. For each experiment, 10 g of Lipozyme TL IM was used as the enzyme bed (apparent bed volume = 19.2 mL). The reaction media were similar to those used for batch acidolysis and interesterification of grapeseed oil with C10:0 or C10 ethyl, respectively. The reaction media were pumped upward from a conical flask placed in a water bath at 40 °C with a peristaltic pump to avoid bed compaction and clogging, as previously described [40]. The bioreactor operated continuously for 150 h for acidolysis and 160 h for interesterification. Periodically, throughout the continuous reactions, 5 mL samples were taken. The collected samples were then frozen at −18 °C until analysis.

The flow rate was 0.5 mL/min, which was calculated as the average value of the ratios between the recovered volumes of reaction media, measured in graduated cylinders, and respective times. A residence time of 13 min was estimated by the ratio between the product of the volume of the enzyme bed by its void fraction corresponding to the volume of reaction medium in the pores of the bed, and the measured flow rate [21]. A void fraction equal to 0.34 was considered for the packed bed with Lipozyme TL IM [40].

Since the apparent steady state was attained only after around four residence times, the start of the continuous operation was considered only after around 50 min of bioreactor operation.

### 3.5. Analysis of Compounds Along Time-Course Reactions

Throughout the acidolysis of the oils with C10:0 or interesterification with C10 ethyl, the initial TAGs, C10:0 or C10 ethyl, and the FFA present in the original oils or released along the reactions, the new TAGs formed, DAGs, and MAGs were quantified by GC in accordance with European Standard EN 14105 [61] with some modifications, as previously described by Souza-Gonçalves et al. [40]. Monononadecanoin (CAS Index Name: Nonadecanoic acid, ester with 1,2,3-propanetriol) was used as the internal standard. A gas chromatograph from Agilent Technologies, Santa Clara, CA, USA, 7820 A (Santa Clara, CA, USA), equipped with an on-column injector, FID, and a capillary column from Agilent J&W DB5-HT (15 m × 0.320 mm ID × 0.10 μm film) was used. Each peak in the chromatograms was identified by comparison with pure standards and with the chromatograms presented in the European Standard EN 14105 [61]. Each compound or group of compounds was quantified as previously described by Souza-Gonçalves et al. [40].

The new TAG yield (%) at time t was calculated as the ratio (in percentage) between the quantification of the new TAG peaks of the chromatogram at time t and the amount of initial TAGs in the chromatogram of the initial blend (time = 0 h). The conversion of the initial TAG at time t was calculated as the ratio (in percentage) between the amount of TAG consumed and the corresponding initial amounts of TAGs. The initial reaction rates were calculated as the slope of the initial straight-line portion of the new TAG production curve along the time course of the reactions (acidolysis or interesterification). The composition of the TAG fraction, corresponding to the maximum yield of new TAGs, was assayed by HPLC, as described for the identification and quantification of the TAG profile in the original oils (c.f. 3.2).

## 4. Conclusions

This study showed that oils obtained from agro-industrial by-products (grapeseed and apricot kernel) and from seeds of a medicinal plant (milk thistle) can be successfully used to produce high-value-added low-calorie dietetic structured lipids using biocatalysis. In the batch interesterification of these oils with C10 Ethyl, catalyzed by Lipozyme RM, in absence of organic solvents, a yield of new TAGs of around 62% was reached for all tested oils, after 6 h of reaction. In batch acidolysis with capric acid, maximum new TAG production was reached after 6, 24, and 30 h reaction with grapeseed (64.7%), milk thistle (56.1%), or apricot kernel (69.7%) oils, respectively.

Both reactions were implemented for grapeseed oil in a continuous packed-bed reactor with up-flow feeding of reaction media using Lipozyme TL IM as a catalyst. Throughout 150 h continuous acidolysis and 160 h continuous interesterification, no lipase deactivation was observed. The average new TAG yields were 79.2% ± 4.1 by interesterification and 61.5% ± 5.91 by acidolysis.

The results obtained are rather promising in terms of scale-up and industrial implementation of these processes to produce low-calorie dietetic TAGs using non-conventional vegetable oils, obtained from agroindustry and agronomic residues, and a stable and non-expensive biocatalyst toward a zero-waste circular bioeconomy.

## Figures and Tables

**Figure 1 molecules-30-01943-f001:**
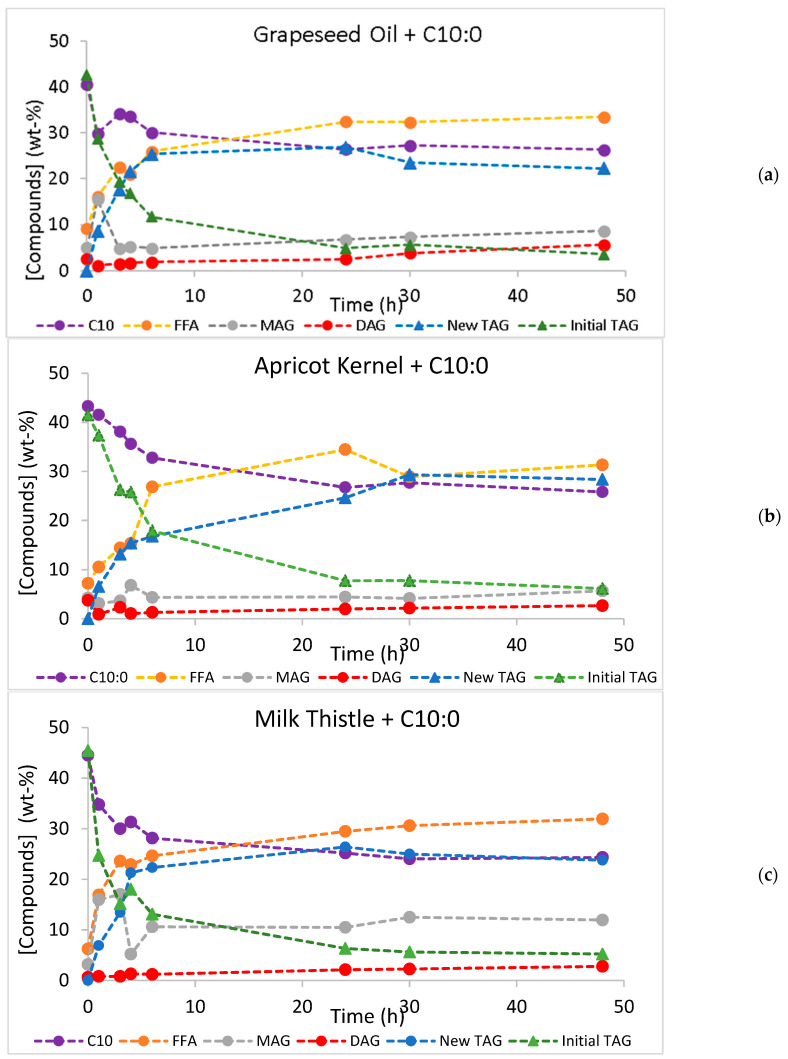
Acidolysis time course of grapeseed (**a**), milk thistle (**b**), and apricot kernel (**c**) oils, with capric acid (C10:0), catalyzed by Lipozyme RM. Standard deviations are not seen in the figures because they were smaller than 0.630, 0.256, and 4.82 for grapeseed, milk thistle, and apricot kernel oil data, respectively.

**Figure 2 molecules-30-01943-f002:**
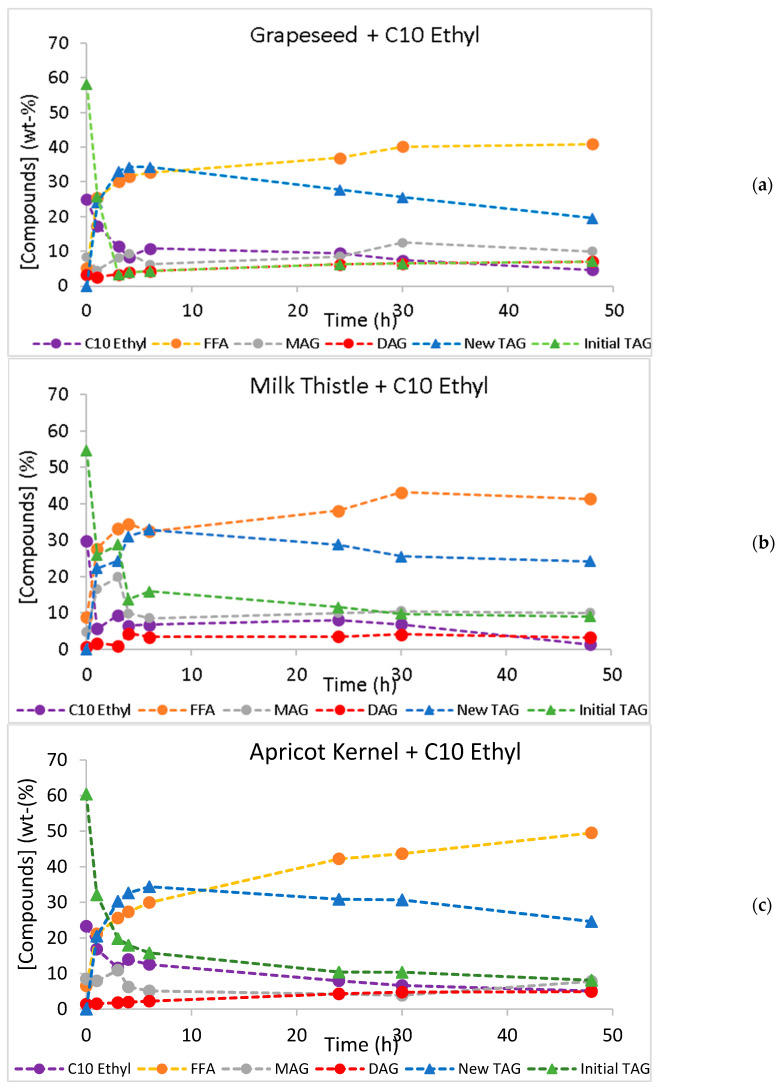
Interesterification time course of grapeseed (**a**), milk thistle (**b**), and apricot kernel (**c**) oils, with ethyl caprate (C10 ethyl) catalyzed by Lipozyme RM. Standard deviations are not seen in the figures, because they were smaller than 0.324, 0.156, and 0.161 for grapeseed, milk thistle, and apricot kernel oils data, respectively.

**Figure 3 molecules-30-01943-f003:**
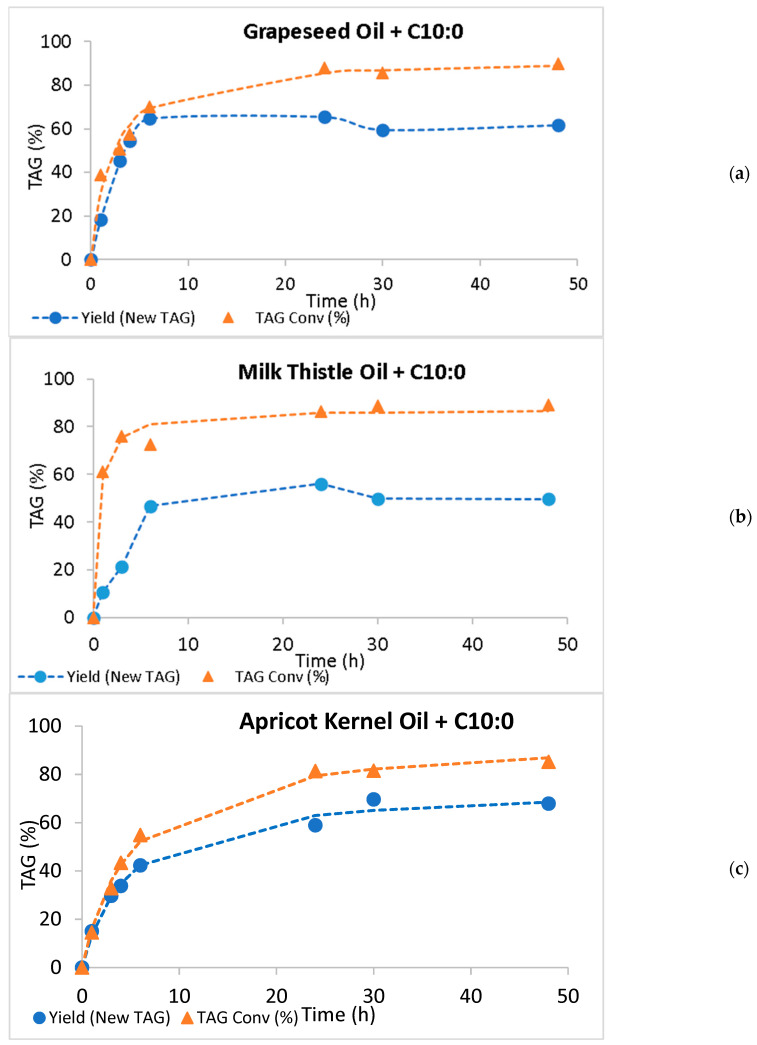
New TAG yield (%) and TAG conversion (%) throughout the 48 h acidolysis of grapeseed (**a**), milk thistle (**b**), and apricot kernel (**c**) oils, with capric acid (C10:0), catalyzed by Lipozyme RM. Standard deviations are not shown in the figures because they were smaller than 1.48, 0.649, and 0.383 for grapeseed, milk thistle, and apricot kernel oil new yield data, and smaller than 1.82, 0.181, 0.765 for grapeseed, milk thistle, and apricot kernel oil TAG conversion data, respectively.

**Figure 4 molecules-30-01943-f004:**
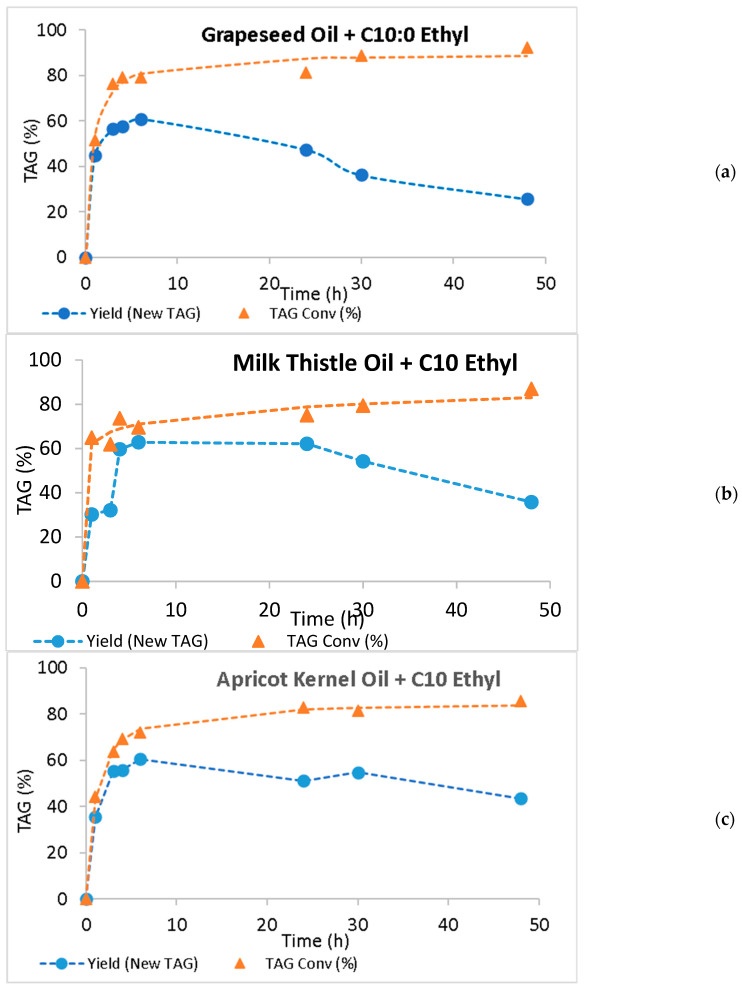
New TAG yield (%) and TAG conversion (%) throughout the 48 h interesterification of grapeseed (**a**)**,** milk thistle (**b**), and apricot kernel (**c**) oils, with ethyl caprate (C10 Ethyl) catalyzed by Lipozyme RM. Standard deviations are not shown in the figures because they were smaller than 0.195, 0.401, and 0.166 for grapeseed, milk thistle, and apricot kernel oil new yield data, and smaller than 0.485, 0.161, and 0.347 for grapeseed, milk thistle, and apricot kernel oil TAG conversion data, respectively.

**Figure 5 molecules-30-01943-f005:**
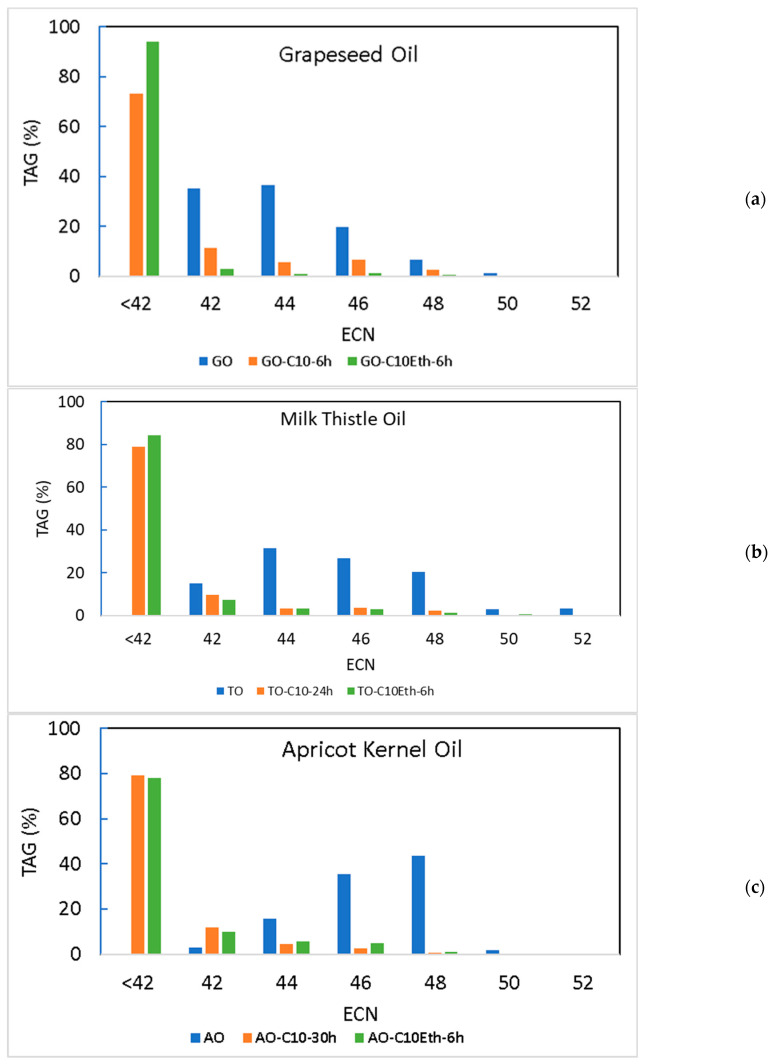
TAG profile (TAGs grouped by ECN) of the initial samples and those corresponding to the maximum yield of dietetic structured lipids (New TAGs): (**a**) GO—grapeseed oil; (**b**) TO—milk thistle oil; (**c**) AO—apricot kernel oil; C10—sample obtained by acidolysis; C10Eth—sample obtained by interesterification.

**Figure 6 molecules-30-01943-f006:**
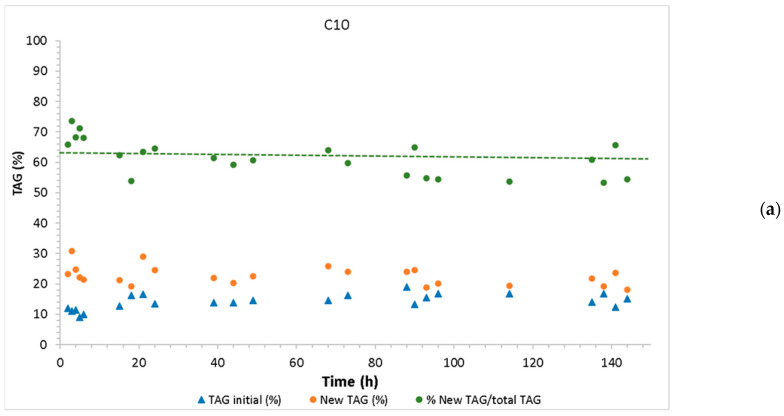
Continuous acidolysis (**a**) and interesterification (**b**) of grapeseed oil, catalyzed by Lipozyme TL IM, presenting the residual initial TAGs, the new TAGs formed, and the yield of new TAGs (yield of new TAGs = % new TAG/total TAGs; dotted lines indicate the average yield value throughout the continuous operation).

**Table 1 molecules-30-01943-t001:** Acid and peroxide values of virgin grapeseed, milk thistle, and apricot kernel oils (average of three determinations ± standard deviation).

Oil	Acid Value (mg KOH/g)	Peroxide Value (meqO_2_/kg)
Grapeseed	0.18 ± 0.00	14.05 ± 0.07
Milk Thistle	1.85 ± 0.00	2.04 ± 0.06
Apricot Kernel	0.979 ± 0.00	1.25 ± 0.01

**Table 2 molecules-30-01943-t002:** Fatty acid composition (%) of virgin grapeseed, milk thistle, and apricot kernel oils (average value of two determinations ± standard deviation; MUFAs—monounsaturated fatty acids; PUFAs—polyunsaturated fatty acids; SFAs—saturated fatty acids).

Fatty Acid	Short Name	Grapeseed Oil	Milk Thistle Oil	Apricot Kernel Oil
Myristic acid	C14:0	0.05 ± 0.002	0.09 ± 0.001	0.02 ± 0.006
Pentadecanoic acid	C15:0	1.0 ± 0.001	0.02 ± 0.000	0.00 ± 0.00
Palmitic acid	C16:0	7.01 ± 0.221	7.71 ± 0.118	5.16 ± 0.059
Palmitoleic acid	C16:1	0.21 ± 0.008	0.10 ± 0.004	0.66 ± 0.006
Heptadecanoic acid	C17:0	0.06 ± 0.002	0.06 ± 0.003	0.04 ± 0.000
Heptadecenoic acid	C17:1	0.03 ± 0.003	0.03 ± 0.002	0.12 ± 0.001
Stearic acid	C18:0	4.15 ± 0.005	5.52 ± 0.037	1.16 ± 0.001
*trans*-Oleic acid	T18:1	0.05 ± 0.006	0.01 ± 0.004	0.02 ± 0.003
Oleic acid	C18:1	19.96 ± 0.066	28.02 ± 0.157	68.61 ± 0.331
*trans*-Linoleic acid	T18:2	0.79 ± 0.103	0.04 ± 0.001	0.03 ± 0.004
Linoleic acid	C18:2	66.62 ± 0.093	51.55 ± 0.061	23.89 ± 0.227
*trans*-Linolenic acid	T18:3	0.07 ± 0.003	0.00 ± 0.000	0.00 ± 0.00
Linolenic acid	C18:3	0.29 ± 0.000	0.17 ± 0.006	0.09 ± 0.006
Arachidic acid	C20:0	0.19 ± 0.000	3.05 ± 0.150	0.1 ± 0.005
Eicosenoic acid	C20:1	0.17 ± 0.005	0.94 ± 0.038	0.1 ± 0.003
Heneicosanoic acid	C21:0	0.12 ± 0.002	0.00 ± 0.00	0.00 ± 0.00
Eicosadienoic acid	C20:2	0.002 ± 0.021	0.00 ± 0.00	0.00 ± 0.00
Behenic acid	C22:0	0.17 ± 0.000	1.88 ± 0.251	0.00 ± 0.00
Erucic acid	C22:1	0.003 ± 0.000	0.03 ± 0.004	0.00 ± 0.00
Lignoceric acid	C24:0	0.06 ± 0.000	0.52 ± 0.113	0.00 ± 0.00
Nervonic acid	C24:1	0.00 ± 0.000	0.10 ± 0.068	0.00 ± 0.00
Σ SFAs		11.82	18.86	6.48
Σ MUFAs		20.42	29.23	69.50
Σ PUFAs		67.76	51.76	24.01

**Table 3 molecules-30-01943-t003:** Main triacylglycerols (TAGs), respective equivalent carbon number (ECNs), and quantities (%) of groups of TAGs with the same ECN (average value of two determinations ± standard deviation), of virgin grapeseed, milk thistle, and apricot kernel oils (L: linoleic; Ln: linolenic; O: oleic; P: palmitic; Po: palmitoleic; S: stearic acids).

ECN	TAGs	Grapeseed Oil	Milk Thistle Oil	Apricot Kernel Oil
42	LLL + (OLLn + PoLL)	34.99 ± 0.023	14.93 ± 0.665	2.50 ± 0.011
44	OLL + (OOLn + PoOL)	23.98 ± 0.102	20.98 ± 0.099	13.49 ± 0.029
44	PLL + PoPoO	11.94 ± 0.314	9.12 ± 0.120	1.47 ± 0.035
46	OOL + LnPP	7.34 ± 0.245	12.07 ± 0.099	27.78 ± 0.031
46	PoOO	7.38 ± 0.028	6.81 ± 0.141	1.26 ± 0.012
46	SLL + PLO	5.188 ± 0.083	6.24 ± 0.042	6.32 ± 0.050
48	PLP	0.38 ± 0.014	7.77 ± 0.106	0.34 ± 0.001
48	OOO + PoPP	1.99 ± 0.030	4.91 ± 0.071	35.24 ± 0.038
48	SOL	2.35 ± 0.072	0.69 ± 0.019	0.98 ± 0.005
48	POO	1.23 ± 0.048	2.17 ± 0.056	6.43 ± 0.021
48	POP	0.54 ± 0.001	2.02 ± 0.020	0.41 ± 0.002
	Σ ECN42	35.13	15.19	2.79
	Σ ECN44	36.67	31.52	15.81
	Σ ECN46	19.95	26.82	35.61
	Σ ECN48	7.01	19.15	43.47
	Σ ECN50	1.00	4.03	1.97
	Σ ECN52	0.24	3.25	0.30

**Table 4 molecules-30-01943-t004:** Maximum yield in new TAGs and TAG conversion (average values ± standard deviation) obtained in each reaction system, time to attain these values, and initial rates (AO—apricot kernel oi; GO—grapeseed oil; TO—milk thistle oil).

System	New TAG Yield Maximum Value (%)	Time (h)	TAG Conversion Maximum Value (%)	Time (h)	Initial Rate (% New TAGs/h)
GO + C10:0	64.7 ± 1.37	6	90.0 ± 0.060	48	13.6
GO + C10 Ethyl	61.0 ± 0.147	6	92.2 ± 0.219	48	44.8
TO + C10:0	56.1 ± 0.117	24	89.1 ± 0.056	48	7.14
TO + C10 Ethyl	62.8 ± 0.176	6	87.0 ± 0.010	48	30.2
AO + C10:0	69.7 ± 0.383	30	85.2 ± 0.084	48	9.9
AO + C10 Ethyl	60.6 ± 0.166	6	85.7 ± 0.035	48	35.5

**Table 5 molecules-30-01943-t005:** Batch production of low-calorie SL by acidolysis or interesterification of several oils with capric acid or C10 ethyl in solvent-free media, catalyzed by different biocatalysts: examples of best reaction conditions and results (crude OP oil: crude olive pomace oil; PBR: packed-bed reactor; ROL: *Rhizopus oryzae* lipase; ROL-MNP: *Rhizopus oryzae* lipase immobilized in magnetic nanoparticles; rROL: recombinant *Rhizopus oryzae* lipase; SCG: spent coffee grounds).

Substrates	Biocatalyst	Best Reaction Conditions	Best Results	Ref.
Acidolysis				
Olive oil + C10:0	Lipozyme RM IM Lipozyme TL IM Novozym 435	Enzyme load: 5% (*w*/*w*) MR = 1:2 Time: 24 h T: 45 °C	Incorporation: RM IM: 27.1 mol%; TL IM: 28.8 mol% Novozym 435: 30.4 mol%	[23]
Grapeseed oil + C10:0	rROL in Amberlite IRA 96 Self-immobilized *C. papaya* lipase	Enzyme load: 5% (*w*/*w*) MR = 1:2 Time: 48 h T: 40 °C	New TAG Yield: rROL: 54.3% *C. papaya lipase*: 38.2%	[25]
Grapeseed oil + C10:0	Lipozyme RM IM Lipozyme TL IM	Enzyme load: 5% (*w*/*w*) MR = 1:2 Time: 24 h T: 45 °C	Incorporation: RM IM: 34.5 mol % TL IM: 27.0 mol %	[27]
Crude OP oil + C10:0 Crude SCG oil + C10:0	Lipozyme TL IM ROL-MNP	Enzyme load: 5% (*w*/*w*) MR = 1:2 T: 50 °C	New TAG Yield: Lipozyme TL IM (48 h): Crude OP oil: ~40% Crude SCG oil: ~48% ROL-MNP (3–5 h): Crude OP oil: ~50% Crude SCG oil: ~50%	[29]
Crude SCG oil + C10:0	Lipozyme RM IM Lipozyme TL IM	Enzyme load: 5% (*w*/*w*) MR = 1:2 T: 50 °C	New TAG Yield: RM IM: ~70% (7 h) TL IM ~50% (48 h)	[30]
Pumpkin oil + C10:0	Lipozyme RM IM	Enzyme load: 5% (*w*/*w*) MR = 1:9 Time: 24 h T: 60 °C	Incorporation: 48.4 mol%	[47]
Grapeseed oil + C10:0	Lipozyme RM IM Lipozyme 435	Enzyme load: 5% (*w*/*w*) MR = 1:3 Time: 12 h T: 65 °C	Incorporation: RM IM: 60.08 mol% Lipozyme 435: 50.78 mol%;	[49]
Sacha inchi oil + C10:0	*Rhizopus oryzae* lipase immobilized in corn cob powder	Bach PBR with recirculation: MR = 1:3 Time: 120 h T: 45 °C	Incorporation: 36 mol%	[50]
Interesterification				
Crude OP oil + C10 Ethyl Crude SCG oil + C10 Ethyl	Lipozyme TL IM ROL-MNP	Enzyme load: 5% (*w*/*w*) MR = 1:2 T: 50 °C	New TAG Yield: Lipozyme TL IM (7 h): Crude OP oil: ~52% Crude SCG oil: ~65% ROL-MNP (3–5 h): Crude OP oil: 46% Crude SCG oil: 26%	[29]
Crude SCG oil + C10 Ethyl	Lipozyme RM IM	Enzyme load: 5% (*w*/*w*) MR = 1:2 Time: 7 h T: 50 °C	New TAG Yield: ~70%	[30]

## Data Availability

The original contributions presented in this study are included in this article. Further inquiries can be directed to the corresponding author.

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
