# Peer review of "Batch and Continuous Lipase-Catalyzed Production of Dietetic Structured Lipids from Milk Thistle, Grapeseed, and Apricot Kernel Oils"

_molecules, 2025, doi:10.3390/molecules30091943_

Round 1
Reviewer 1 Report
Comments and Suggestions for Authors
The study provides a comprehensive analysis of the production of dietetic structured lipids (SLs) from non-conventional oils using lipase-catalyzed acidolysis and interesterification, contributing to sustainable and zero-waste bioeconomy practices. The comparison between batch and continuous bioreactor performance, along with detailed oil characterization, enhances the understanding of the production process. However, several weaknesses should be addressed before the manuscript can be considered for publication:
1- Results should be consistently expressed as “average ± standard deviation” in tables (see Tables 2 and 3).
2- Error bars should be included in the figures to improve data visualization and interpretation.
3- The authors state that “the highest rates of batch acidolysis and interesterification reactions were observed with grapeseed oil.” To strengthen the discussion, a clearer explanation of why grapeseed oil performed best should be provided.
4- The manuscript references numerous studies on batch and continuous reactors for comparison. To enhance readability and interpretation, I suggest summarizing these studies in a table, highlighting the advantages of the present study.
5- The variation in reaction times required to achieve maximum conversion is not adequately discussed. The authors should provide more insight into why reaction kinetics differ between acidolysis and interesterification.
6- Line 431-432: "The biocatalyst maintained its activity either in acidolysis or interesterification" Suggested revision: "The biocatalyst maintained its activity in both acidolysis and interesterification."
Comments on the Quality of English Language
none
Author Response
First of all, we would like to thank you for the time and valuable suggestions which greatly helped to improve our manuscript entitled “Batch and continuous lipase-catalyzed production of dietetic structured lipids from milk thistle, grapeseed and apricot kernel oils”, submitted to be considered for publication in Journal Molecules as a research article.
All the modifications in the original manuscript are in red.
We hope all the questions were answered accordingly and this version of the manuscript will meet the required standards for publication in Molecules.
Best regards,
Suzana Ferreira-Dias
Reviewer 1
Comments and Suggestions for Authors
The study provides a comprehensive analysis of the production of dietetic structured lipids (SLs) from non-conventional oils using lipase-catalyzed acidolysis and interesterification, contributing to sustainable and zero-waste bioeconomy practices. The comparison between batch and continuous bioreactor performance, along with detailed oil characterization, enhances the understanding of the production process. However, several weaknesses should be addressed before the manuscript can be considered for publication:
1- Results should be consistently expressed as “average ± standard deviation” in tables (see Tables 2 and 3).
Ans: You are right. We had forgotten to add the standard deviation values. The tables were corrected as requested.
2- Error bars should be included in the figures to improve data visualization and interpretation.
Ans: As said in the first version of the manuscript, the standard deviation (STD) values are very small and could not be seen in the figures. Therefore, we decided to add this information and the highest STD values in the legends of the figures 1 to 4. For example, in Fig 1 is written: “Standard deviations are not seen in the figures because they were smaller than 0.630, 0.256, and 4.82 for grapeseed, milk thistle and apricot kernel oils data, respectively”.
If instead of STD values, we use error values, these will be even smaller than the STD values because the standard error of the mean is calculated by dividing the standard deviation by the square root of the number of elements of the sample. Please see the link: https://www.scribbr.com/statistics/standard-error/.
3- The authors state that “the highest rates of batch acidolysis and interesterification reactions were observed with grapeseed oil.” To strengthen the discussion, a clearer explanation of why grapeseed oil performed best should be provided.
Ans: As said in the first version of the manuscript we suggest that “the best results with grapeseed oil may be explained a slight preference of Lipozyme RM towards oils richer in linoleic acid (66.62 % in grapeseed oil and 51.55 % in milk thistle oil) than in oils richer in oleic acid (68.61 % in apricot kernel oil).
The highest reaction rates observed for acidolysis and interesterification of grapeseed oil were the reason for choosing this oil in continuous bioreactor. High reaction rates are important in continuous process implementation because they allow lower residence times to attain high conversions and yields. This information was added to the section “2.3. Continuous Production of Low-calorie TAGs”
Moreover, the method used for the calculation of the initial reaction rates was added to the section “ 3.5. Analysis of compounds along time-course reactions”, as follows: “The initial reaction rates were calculated as the slope of the initial straight-line portion of the New TAG production curve along time-course reactions (acidolysis or interesterification)”.
4- The manuscript references numerous studies on batch and continuous reactors for comparison. To enhance readability and interpretation, I suggest summarizing these studies in a table, highlighting the advantages of the present study.
Ans: Thank you very much for the good suggestion. A new table containing some specific examples was added to facilitate the comparison of our results with those from similar studies (Table 5: Batch production of low-calorie SL by acidolysis or interesterification of several oils with capric acid or C10 Ethyl, in solvent free media, catalyzed by different biocatalysts: examples of best reaction conditions and results).
5- The variation in reaction times required to achieve maximum conversion is not adequately discussed. The authors should provide more insight into why reaction kinetics differ between acidolysis and interesterification.
Ans: The following information was added to the manuscript in section “2.2. Batch Production of Low-calorie TAGs”:
“Acidolysis consists of a reaction between an ester (TAG) and free fatty acids, which will replace the original fatty acid acids in the TAG. When sn-1,3 regioselective lipases are used, this exchange will occur at positions sn-1,3. The interesterification consists of an ester-ester interchange being, in the present study, between the oil TAGs and capric acid ethyl ester.
Both acidolysis and interesterification reactions are usually considered as a two-step reaction, consisting of the hydrolysis of ester bonds in the TAGs followed by the esterification of acyl groups in glycerol backbone. The hydrolysis of a TAG results in a diacylglycerol (DAG) and the release of a fatty acid molecule. The formed DAG can esterify with another FFA to form a new TAG or to be hydrolyzed into a monoacylglycerol (MAG) and the release of a FFA. Diacylglycerols and monoacylglycerols are thermodynamically unstable compounds, acting as reaction intermediates [Xu, 2003].
The rate of DAG formation increases with the water content in the system [Xu, 2003]. For the first hydrolytic step, the presence of water is required. According to Heisler et al. [1991], in the isomerization of 1,2-dipalmitin into the 1,3-isomer, the trace water present in lyophilized sn-1,3 regioselective lipase powder was used in the first step of DAG hydrolyses. These authors also observed that the hydrolytic step was faster than the esterification step.
Therefore, the control of the water content in the system is very important since, to achieve high yields of New TAGs, low contents of DAGs, MAGs and FFAs are required at the equilibrium. It means that after the hydrolytic initial step, the esterification reaction must occur at higher rate than hydrolysis. The reaction equilibrium depends on the extent of hydrolysis versus esterification steps and the optimal amount of water required mainly depends on the type of lipase used [Xu, 2003].”
6- Line 431-432: "The biocatalyst maintained its activity either in acidolysis or interesterification" Suggested revision: "The biocatalyst maintained its activity in both acidolysis and interesterification."
Ans: done.
Comments on the Quality of English Language
none
Reviewer 2 Report
Comments and Suggestions for Authors
The manuscript by Ş. Akbaş describes the development of a process to produce low-calorie triacylglycerols (TAGs) using immobilized lipase from by-product oils. The authors demonstrate that low-calorie TAGs can be produced from three types of oils (milk thistle, grapeseed, and apricot kernel) using two reaction strategies (acidolysis and interesterification) in both batch and continuous processes. This research provides important information for developing processes that convert industrial by-products into more valuable compounds.
1) The introduction effectively explains the necessity of upcycling industrial by-products, the industrial demand for low-calorie TAGs, and production strategies, successfully capturing reader interest. However, all three results sections are disappointing. They primarily list previous research findings with minimal connection to the current work's results. This approach fails to connect with the discussion of the current findings and significantly discourages reader's attention. Consequently, I cannot determine whether this manuscript is intended as an article or a review, leaving me confused about the authors' central message.
2) If the goal of listing previous studies is to compare current data with previous findings, I suggest creating a comparative table that clearly highlights the differences between prior research and the current work.
3) Has the detailed profile of the produced TAGs been analyzed? Figure 5 introduces this but merely classifies based on ECN (Equivalent Carbon Number) without mentioning specific TAGs as in Table 3. The discussion should address which ECN values are industrially significant from a dietetic perspective and which were specifically targeted by the authors.
4) Section 2.2 indicates experiments were conducted under solvent-free conditions. Under such conditions, particularly for interesterification reactions, lipase-mediated hydrolysis would not be expected to be favorable. Yet, the authors mention TAG yield decreases due to hydrolysis during the reaction. What explains this phenomenon?
5) In Figure 1, the concentration of C10 substrate decreased only about 10-15% from an initial 40%, meaning most C10 did not participate in the reaction. With an oil donor mole ratio of 1:2, wouldn't most produced TAGs be MLL type? Has the ratio of MLL to MLM types been analyzed?
6) For all reactions, how does the detailed composition of new TAGs change with reaction time? Does the ratio of MLL to MLM types change continuously with reaction time regardless of yield? If so, what is the trend?
7) What are the specific differences in TAG composition after interesterification versus acidolysis? Beyond ECN, what differences exist in MLL and MLM compositions?
8) Figures 3 and 4 appear to be yield and conversion graphs for the results in Figures 1 and 2. I suggest merging these by adding them as subsections (d, e, f) to the existing figures.
9) In Figure 5, the reaction times for acidolysis and esterification differ for all three oils, making interpretation difficult. Note that these times also differ from those for maximum yield and conversion. Is there a specific reason for selecting data from these different time points?
10) Is there a particular reason for using lipase TL in the continuous reaction? Why different lipases were used for the batch and continuous reactions?
11) Abbreviations should be defined at their first appearance in the document. For example, ECN first appears on line 168, but its definition appears in Table 3 (line 187), making readers move back and forth to understand the content.
12) The number of references is excessive, with 15 self-citations out of 66 total references. Please limit citations to only those necessary for this manuscript and reduce the high proportion of self-citations.
13) I believe TO-C10-30h in Figure 5 is a typo of TO-C10Eth-30h.
Author Response
First of all, we would like to thank you for the time and valuable suggestions which greatly helped to improve our manuscript entitled “Batch and continuous lipase-catalyzed production of dietetic structured lipids from milk thistle, grapeseed and apricot kernel oils”, submitted to be considered for publication in Journal Molecules as a research article.
All the modifications in the original manuscript are in red.
We hope all the questions were answered accordingly and this version of the manuscript will meet the required standards for publication in Molecules.
Best regards,
Suzana Ferreira-Dias
Reviewer 2
Comments and Suggestions for Authors
The manuscript by Ş. Akbaş describes the development of a process to produce low-calorie triacylglycerols (TAGs) using immobilized lipase from by-product oils. The authors demonstrate that low-calorie TAGs can be produced from three types of oils (milk thistle, grapeseed, and apricot kernel) using two reaction strategies (acidolysis and interesterification) in both batch and continuous processes. This research provides important information for developing processes that convert industrial by-products into more valuable compounds.
1) The introduction effectively explains the necessity of upcycling industrial by-products, the industrial demand for low-calorie TAGs, and production strategies, successfully capturing reader interest. However, all three results sections are disappointing. They primarily list previous research findings with minimal connection to the current work's results. This approach fails to connect with the discussion of the current findings and significantly discourages reader's attention. Consequently, I cannot determine whether this manuscript is intended as an article or a review, leaving me confused about the authors' central message.
Ans: We tried to address your concerns about the discussion section. However, we would like to have more specific clues to improve it.
2) If the goal of listing previous studies is to compare current data with previous findings, I suggest creating a comparative table that clearly highlights the differences between prior research and the current work.
Ans: Thank you very much for the good suggestion. A new table containing some specific examples was added to facilitate the comparison of our results with those from similar studies (Table 5: Batch production of low-calorie SL by acidolysis or interesterification of several oils with capric acid or C10 Ethyl, in solvent free media, catalyzed by different biocatalysts: examples of best reaction conditions and results).
3) Has the detailed profile of the produced TAGs been analyzed? Figure 5 introduces this but merely classifies based on ECN (Equivalent Carbon Number) without mentioning specific TAGs as in Table 3. The discussion should address which ECN values are industrially significant from a dietetic perspective and which were specifically targeted by the authors.
Ans: The exact identification of the new TAGs was not possible with the equipment we have in our laboratory. We contacted several colleagues from abroad to see if it was possible in their labs this identification. However, none of them could do it because it is not a straight forward task, since it is hard to find available standards of these new TAGs in the market and/or HPLC-MS was needed.
Under the conditions used in our laboratory (recognized by the International Olive Council), we can only identify TAGs with ECN equal to 42 (e.g. LLL) or higher (by comparison with TAG standards and standard chromatograms of edible oils). Therefore, we decided to quantify all the New TAGs as a whole, only knowing that they have ECN < 42.
From a dietetic perspective, the presence of at least one medium-chain FA in TAG is beneficial. Thus, considering that:
Main TAGs in grapeseed oil and milk thistle oils are LLL; OLL and OOL;
Main TAGs in apricot kernel oil main TAGs: OOO, OOL, OLL;
The main New TAGs (one or two substitutions) will be as follows:
LLC (ECN = 38); CLC (ECN = 34); OLC (ECN = 40; COO (ECN = 42); COC (ECN = 36)
This information was added in the manuscript as follows: “The main New TAGs, containing one or two capric acid, C, residues (mainly at positions sn-1,3 but some at position sn-2 due to eventual acyl migration phenomenon) will be: CLC (ECN = 34); COC (ECN = 36); LLC (ECN = 38); OLC (ECN =40); OOC (ECN = 42)”.
4) Section 2.2 indicates experiments were conducted under solvent-free conditions. Under such conditions, particularly for interesterification reactions, lipase-mediated hydrolysis would not be expected to be favorable. Yet, the authors mention TAG yield decreases due to hydrolysis during the reaction. What explains this phenomenon?
Ans: The mechanism of lipase-catalyzed acidolysis and interesterification was added to this version of the manuscript. As a matter of fact, both acidolysis and interesterification reactions are usually considered as a two-step reaction, consisting of the hydrolysis of ester bonds in the TAGs followed by the esterification of acyl groups in glycerol backbone. The studies on these reactions show that the first hydrolytic step occurs even under a very low water content of the reaction medium, in solvent-free systems. The trace water existing in the immobilized biocatalysts and in the oils are enough to promote this hydrolysis.
The following text added to the manuscript (2.2. Batch Production of Low-calorie TAGs by Acidolysis and Interesterification):
“Acidolysis consists of a reaction between an ester (TAG) and free fatty acids, which will replace the original fatty acids in the TAG. When sn-1,3 regioselective lipases are used, this exchange will occur at positions sn-1,3. The interesterification consists of an ester-ester interchange being, in the present study, between the oil TAGs and capric acid ethyl ester.
Both acidolysis and interesterification reactions are usually considered as a two-step reaction, consisting of the hydrolysis of ester bonds in the TAGs followed by the esterification of acyl groups in glycerol backbone. The hydrolysis of a TAG results in a diacylglycerol (DAG) and the release of a fatty acid molecule. The formed DAG can esterify with another FFA to form a new TAG or to be hydrolyzed into a monoacylglycerol (MAG) and the release of a fatty acid molecule. Diacylglycerols and monoacylglycerols are thermodynamically unstable compounds, acting as reaction intermediates [Xu, 2003].
The rate of DAG formation increases with the water content in the system [Xu, 2003]. For the first hydrolytic step, the presence of water is required. According to Heisler et al. [1991], in the isomerization of 1,2-dipalmitin into the 1,3-isomer, the trace water present in lyophilized sn-1,3 regioselective lipase powder was used in the first step of DAG hydrolyses. These authors also observed that the hydrolytic step was faster than the esterification step.
Therefore, the control of the water content in the system is very important since, to achieve high yields of New TAGs, low contents of DAGs, MAGs and FFAs are required at the equilibrium. It means that after the hydrolytic initial step, the esterification reaction must occur at higher rate than hydrolysis. The reaction equilibrium depends on the extent of hydrolysis versus esterification steps and the optimal amount of water required mainly depends on the type of lipase used [Xu, 2003].”
Yet, the authors mention TAG yield decreases due to hydrolysis during the reaction. What explains this phenomenon?
Ans: This decrease is observed because the hydrolytic step was faster than the subsequent esterification step. This explanation was already presented in the first version of the manuscript, as follows:
“High differences between TAG conversion and New TAG yield suggest that the hydrolytic step is faster than the esterification reaction, which results in lower production of New TAGs and increase in MAGs, DAGs and FFAs in the reaction medium.”
5) In Figure 1, the concentration of C10 substrate decreased only about 10-15% from an initial 40%, meaning most C10 did not participate in the reaction. With an oil donor mole ratio of 1:2, wouldn't most produced TAGs be MLL type? Has the ratio of MLL to MLM types been analyzed?
Ans: We could not determine the ratio of MLL to MLM, due to analytical constraints, as previously explained. We only could analyze the TAG fraction in terms of % of each peak and ECN. From the chromatograms, we saw that almost all the initial TAGs were consumed and transformed into New TAG species but we cannot know if one or two substitutions of original FA by capric acid occurred.
You are right about the Figures 1 and 2. However, the amount of each compound is presented in wt% of the total mass of reagents, and not in mol% (we do not know the molecular weight of the new species formed). Since the molecular weight for capric acid is 172.26 g/mol, for triolein is 885.4 g/mol and for trilinolein is 879.4 g/mol, it is not easy to know if one or two substitutions of original FA by capric acid occurred in the new TAGs.
6) For all reactions, how does the detailed composition of new TAGs change with reaction time? Does the ratio of MLL to MLM types change continuously with reaction time regardless of yield? If so, what is the trend?
Ans: We could not get this information due to analytical constraints, as previously explained. In the GC chromatograms of the time-course, we can see the peaks of original TAGs diminishing, while new peaks in the TAG region (new TAGs), DAGs, MAGs and FFA released from the original TAGs increased. This trend is observed in the time-courses of Figures 1 and 2.
7) What are the specific differences in TAG composition after interesterification versus acidolysis? Beyond ECN, what differences exist in MLL and MLM compositions?
Ans: We could not get this information due to analytical constraints, as previously explained.
8) Figures 3 and 4 appear to be yield and conversion graphs for the results in Figures 1 and 2. I suggest merging these by adding them as subsections (d, e, f) to the existing figures.
Ans: Thank you for the suggestion. However, we tried but the figures became rather small, and we decided to keep them as in the first version of the manuscript.
9) In Figure 5, the reaction times for acidolysis and esterification differ for all three oils, making interpretation difficult. Note that these times also differ from those for maximum yield and conversion. Is there a specific reason for selecting data from these different time points?
Ans: As explained in the first version of the manuscript, we decided to analyse the TAG composition of the samples with the highest yields of New TAGs (Table 4). Since the maximum yield was not attained at the same time for the different systems, the results presented correspond to different reaction times. Maximum yield is not always coincident with maximum conversion because, if the hydrolytic step is faster than the esterification step, the hydrolysis of TAGs may occur with decrease in New TAG yield but an increase in conversion due to the formation of partial acylglycerols and fatty acid release. This information was already in the first version of the manuscript.
10) Is there a particular reason for using lipase TL in the continuous reaction? Why different lipases were used for the batch and continuous reactions?
Ans: Yes. As previously explained in (lines 453-456), “Instead of Lipozyme RM, Lipozyme TL IM was chosen. This biocatalyst is very sensitive to shear stress, which makes it not suitable for batch reactions under magnetic stirring but has a much lower price than Lipozyme RM (about 8-fold cheaper, according to the manufacturer).”
Lipozyme TL IM is not adequate to batch reactions under magnetic stirring because the immobilization particles are rapidly destroyed. Due to this constraint, we decided to use Lipozyme RM IM in batch reactions. In previous studies of our group, both Lipozyme RM and TL IM performed well in continuous bioreactors. Due to enzyme costs, we decided to test only Lipozyme TL IM in the continuous bioreactor.
11) Abbreviations should be defined at their first appearance in the document. For example, ECN first appears on line 168, but its definition appears in Table 3 (line 187), making readers move back and forth to understand the content.
Ans: Thank you. The meaning of ECN was moved to the place of its first appearance.
12) The number of references is excessive, with 15 self-citations out of 66 total references. Please limit citations to only those necessary for this manuscript and reduce the high proportion of self-citations.
Ans: Sorry but we do not think that 15 self-citations in 66 references is too much because both the corresponding author and Prof. Natália Osório work in Structured lipids production for several decades. We only used the citations that we believe were important for this study.
13) I believe TO-C10-30h in Figure 5 is a typo of TO-C10Eth-30h.
Ans: In fact, the legends of the series for milk thistle oil were not correct. It is correct now. Thank you!
Round 2
Reviewer 1 Report
Comments and Suggestions for Authors
The manuscript has been improved a lot after revision.
Author Response
We would like to thank you for the evaluation of the revised version of our manuscript entitled “Batch and continuous lipase-catalyzed production of dietetic structured lipids from milk thistle, grapeseed and apricot kernel oils”, submitted to be considered for publication in Journal Molecules as a research article.
Best regards,
Suzana Ferreira-Dias
Reviewer 2 Report
Comments and Suggestions for Authors
The authors have adequately addressed the majority of the comments raised in the first review. The manuscript is now substantially improved. However, some issues remain that need to be clarified or revised prior to publication.
1) Sections such as lines 408–428 and 492–556 contain extensive details on previous studies without offering sufficient discussion or comparison to the current findings. This weakens the focus of the manuscript and may confuse the reader about the core results in this study. Please revise these sections to enhance relevance and ensure they support the main results.
2) Some parts of the manuscript are unclear for inclusion. For instance, the content in lines 406–407 looks loosely connected to the main content.
3) The proportion of self-citations remains relatively high. While prior work from the authors may be foundational, please provide specific justification for citing at least references 17, 19, 20, 27, 28, and 29.
Author Response
First of all, we would like to thank you for the time and valuable suggestions which greatly helped to improve our manuscript entitled “Batch and continuous lipase-catalyzed production of dietetic structured lipids from milk thistle, grapeseed and apricot kernel oils”, submitted to be considered for publication in Journal Molecules as a research article.
All the modifications in the original manuscript are in red.
We hope all the questions were answered accordingly and this version of the manuscript will meet the required standards for publication in Molecules.
Best regards,
Suzana Ferreira-Dias
Reviewer 2
Comments and Suggestions for Authors
The authors have adequately addressed the majority of the comments raised in the first review. The manuscript is now substantially improved. However, some issues remain that need to be clarified or revised prior to publication.
1) Sections such as lines 408–428 and 492–556 contain extensive details on previous studies without offering sufficient discussion or comparison to the current findings. This weakens the focus of the manuscript and may confuse the reader about the core results in this study. Please revise these sections to enhance relevance and ensure they support the main results.
Ans: Thank you for the suggestions. In fact, the lines 408–428 concern the discussion of the results presented in Table 5. Therefore, as you suggested, we shortened the text as follows (lines 406-417):
“In batch reactions, Lipozyme TL IM preferred interesterification with C10 Ethyl than acidolysis with capric acid, using crude spent coffee ground (SCG) and crude olive pom-ace (OP) oils, in solvent-free medium (Table 5) [29]. However, when Rhizopus oryzae lipase immobilized in magnetic nanoparticles (ROL-MNP) was used, the acidolysis reaction was faster than interesterification (Table 5) [29].
Mota et al. [30] used crude SCG oil to produce MLM, either by acidolysis with caprylic acid or capric acid or by interesterification with ethyl caprylate (C8 Ethyl) or C10 Ethyl, catalyzed by Lipozyme RM IM or Lipozyme TL IM (Table 5). With this oil, higher yields in new TAGs were obtained by acidolysis, when Lipozyme RM IM was used, or by interesterification, with Lipozyme TL IM [30].
In the present study, where Lipozyme RM was used in batch reactions, interesterification was faster than acidolysis, but the New TAGs yield was more dependent on the oil used than on the reaction followed.”
In the section about the operational stability, we decided to maintain only the examples where Lipozyme TL IM was used, as follows (lines 485-514):
“In batch acidolysis, Lipozyme TL IM showed a first-order deactivation kinetics of virgin olive oil of capric (47.2 h) acids [23]. When used in a packed-bed reactor with continuous recirculation for the interesterification of soybean oil with TAGs rich in caprylic and capric acids, no significant deactivation of this biocatalyst was observed after 25 consecutive batches (total of 6.7 h) [28]. The operational stability exhibited by Lipozyme TL IM, in batch acidolysis reuses, is lower than that observed for this biocatalyst in present study, using a continuous PBR.
In continuous interesterification in a packed bed reactor, between fish oil and medium-chain TAGs, no deactivation of Lipozyme TL IM was observed along 2-week continuous operation [22].
Souza-Gonçalves et al. [40] performed the acidolysis of high acidic (12-29 % FFA) crude olive pomace oils with C8:0 or C10:0 acids and interesterification with their ethyl ester forms, catalyzed by Lipozyme TL IM or Lipozyme RM IM, in continuous PBR, for 70 h, in solvent-free media at 40 °C. Along the continuous operation, no biocatalyst deactivation was observed, except for Lipozyme TL IM in the acidolysis with capric acid (linear deactivation; half-life time = 228 h) and Lipozyme RM IM in the interesterification with ethyl caprylate (first-order deactivation; half-life time = 74 h).
The high stability of Lipozyme TL IM observed in our study, either in acidolysis of grapeseed oil with capric acid or interesterification with C10 Ethyl, is comparable to the results previously obtained with this biocatalyst when used in continuous bioreactors [22, 40].”
2) Some parts of the manuscript are unclear for inclusion. For instance, the content in lines 406–407 looks loosely connected to the main content.
Ans: This example, as well as the reaction conditions and results, are described in Table 5. Since the discussion of these results were in lines 434-445 (version 2 of the manuscript, now lines 430-433), we decided to remove the sentence, as you suggested.
3) The proportion of self-citations remains relatively high. While prior work from the authors may be foundational, please provide specific justification for citing at least references 17, 19, 20, 27, 28, and 29.
Ans: To decrease the number of self-citations, we removed the suggested references.